# Translating Molecular Biology Discoveries to Develop Targeted Cancer Interception in Barrett’s Esophagus

**DOI:** 10.3390/ijms241411318

**Published:** 2023-07-11

**Authors:** Sohini Samaddar, Daniel Buckles, Souvik Saha, Qiuyang Zhang, Ajay Bansal

**Affiliations:** 1Department of Internal Medicine, University of Kansas Health System, Kansas City, KS 66160, USA; ssamaddar@kumc.edu (S.S.); souvik.saha14@gmail.com (S.S.); 2Department of Gastroenterology and Hepatology, University of Kansas Health System, Kansas City, KS 66160, USA; dbuckles@kumc.edu; 3Center for Esophageal Diseases, Department of Medicine, Baylor University Medical Center, Dallas, TX 75246, USA; qiuyang.zhang@bswhealth.org; 4Center for Esophageal Research, Baylor Scott & White Research Institute, Dallas, TX 75246, USA; 5University of Kansas Cancer Center, Kansas City, KS 66160, USA

**Keywords:** esophageal adenocarcinoma, chemoprevention, Barrett’s esophagus

## Abstract

Esophageal adenocarcinoma (EAC) is a rapidly increasing lethal tumor. It commonly arises from a metaplastic segment known as Barrett’s esophagus (BE), which delineates the at-risk population. Ample research has elucidated the pathogenesis of BE and its progression from metaplasia to invasive carcinoma; and multiple molecular pathways have been implicated in this process, presenting several points of cancer interception. Here, we explore the mechanisms of action of various agents, including proton pump inhibitors, non-steroidal anti-inflammatory drugs, metformin, and statins, and explain their roles in cancer interception. Data from the recent AspECT trial are discussed to determine how viable a multipronged approach to cancer chemoprevention would be. Further, novel concepts, such as the repurposing of chemotherapeutic drugs like dasatinib and the prevention of post-ablation BE recurrence using itraconazole, are discussed.

## 1. Introduction

Esophageal adenocarcinoma (EAC), a lethal tumor, is the fastest-increasing cancer in North America. To make matters worse, EAC is increasing in both men and women under 50 years of age [1]. The overall mean 5-year survival rate for EAC is a somber 18.0% [2]; and 50.0% to 80.0% of patients with EAC present with incurable, locally advanced, unresectable, or metastatic disease [2]. EAC arises from Barrett’s esophagus (BE), which identifies the patient population at risk. In recent years, the pathophysiology of BE has been extensively studied [3,4,5], leading to the identification of novel targets for chemoprevention, though a better term might be cancer interception, which refers to active cancer prevention using targeted immunologic and pharmacologic methods to disrupt preinvasive carcinogenesis [6]. In this article, we discuss the biology of molecular targets for cancer interception in BE. Table 1 lists the clinical trials of cancer interception in BE to inform the audience about the study design, endpoints, and the variety of agents tested thus far.

## 2. Prevention in Native Barrett’s Esophagus

### 2.1. Proton Pump Inhibitors

Acid and bile exposure due to gastroesophageal reflux disease (GERD) is the primary initiator of esophageal injury that leads to the development and progression of BE [7,8,9] (Figure 1). Therefore, proton pump inhibitors have been tested for cancer interception in BE (Figure 1). PPIs can not only reduce acid reflux but can also alleviate bile reflux by decreasing the volume of gastric secretions, thereby reducing the transport of bile acids. Multiple studies have proposed that acid and bile acids work synergistically in BE progression [10,11].

The conventional assumption behind acid- and bile-induced injury was direct chemical injury. However, recent reports suggest that the injury may be inflammatory in nature. Using a mouse model of reflux-induced injury caused by esophagoduodenostomy, the investigators carefully examined serial sections of whole-thickness esophagus and showed that submucosal inflammation by T cells occurred before surface injury [12]. In vitro experiments also revealed that exposure of esophageal squamous cells to acidified bile salts induces the production of inflammatory cytokines IL8 and IL1β, which affected the migratory behavior of T cells and neutrophils [12]. These findings were supported by a human trial (N = 12) involving patients with histopathologically confirmed healed esophagitis, who discontinued PPIs for a short duration [13]. Again, infiltration by T cells preceded the development of clinically evident esophageal injury by approximately two weeks. If the injury was caused by the direct corrosive effect of acid and bile, one would expect the macroscopic injury to occur much earlier after PPI discontinuation. One possible main mediator is hypoxia-inducible factor (HIF) 2 alpha [14]. HIF alpha inhibitors are now readily available; and although this may be far in the future, these drugs could one day play a role in BE cancer interception [15].

So far, despite there being a good biological rationale for the cancer interception effect of PPIs in the progression of BE, clinical data are not decisive. Over the years, meta-analyses and many observational studies have reported inconsistent results regarding the cancer-protective nature of PPIs, including studies in support of [16,17,18] and against [19,20] this hypothesis, and yet with others being inconclusive [21,22]. Recent American Gastroenterological Association guidelines, however, do not recommend de-prescribing PPIs in patients with BE, citing possible chemopreventive effects [23]. AspECT, the largest trial to date on esomeprazole and aspirin in BE, addressed the role of PPIs and is discussed below [24].

### 2.2. Non-Steroidal Anti-Inflammatory Drugs and Aspirin

Non-steroidal anti-inflammatory drugs (NSAIDs), including aspirin, could play a role in cancer interception in BE by reducing inflammatory injury, as discussed earlier. The primary mechanism of action of NSAIDs is proposed to be through the inhibition of COX2 expression (Figure 1 and Figure 2). COX2 expression is associated with higher degrees of dysplasia in BE [25,26,27,28]. The production of prostaglandins, particularly PGE2, is mediated by COX2 expression and can dampen immune responses, promote cellular proliferation, and induce angiogenesis [29]. The COX1/2-driven thromboxane A2 pathway may also facilitate neoplastic progression [30]. Thus, COX2 inhibition by NSAIDs presents an attractive method for cancer interception.

Souza et al. treated EAC cell lines with a COX2 selective inhibitor to demonstrate the suppression of cellular growth [31]. Another study, that exposed ex vivo cultures established from endoscopic biopsies of patients with BE to COX-2 inhibition, showed a reduction in cell proliferation by 55.0% (*p* < 0.001) [32]. Supporting the specificity of COX-2 in this process were observations that exogenous PGE2 reversed the process. The same group of investigators used an animal model of BE [33], i.e., rats who underwent esophagojejunostomies, to test either a selective COX-2 inhibitor (MF-Tricyclic) or a nonselective COX inhibitor (Sulindac). Compared to a placebo, MF-Tricyclic and sulindac decreased the risk of EAC development by 55.0% (*p* < 0.008), and 79.0% (*p* < 0.001), respectively. These phenotypic changes in mice models are consistent with human observations. Using whole exam sequencing, after matching for important covariates like smoking and P53 expression, NSAIDs were shown to affect as many as half of the genes in nine oncogenic pathways [34]. Clinical data seem to be in support of the above observations [35,36,37,38]. In a meta-analysis of 14 case-control and cohort studies [36], NSAIDs were associated with a 36.0% reduction in the risk of EAC (*p* = 0.10) when compared with BE patients. Further, another pooled analysis showed a lower risk of EAC with NSAID use (OR 0.68, 95% CI: 0.56–0.82) [37]. While these initial data are indeed promising, they have not yet resulted in successful trials. A randomized controlled trial of 200 mg BID of celecoxib vs. a placebo for 48 weeks in patients with BE with either low- or high-grade dysplasia did not reduce the incidence of dysplasia, a surrogate endpoint of EAC [39]. However, celecoxib is a selective COX-2 inhibitor and could have different effects compared with non-selective NSAIDs.

There has been extensive interest in understanding the differences between aspirin and NSAIDs. Aspirin is an irreversible inhibitor of COX enzymes, vs. non-aspirin NSAIDs which are reversible inhibitors. Additionally, aspirin has the distinctive property of inhibiting the inflammatory NFκB and Wnt signaling pathways [40,41] (Figure 1 and Figure 2). Therefore, even though both belong under the general umbrella of NSAIDs, they could have different cancer interception effects. In the meta-analysis discussed above [36], the use of aspirin was associated with a 27.0% reduction in the risk of EAC (0.73 95% CI: 0.65–0.83). The relative benefits of aspirin vs. non-aspirin NSAIDs need to be further understood. The AsPECT trial addressed aspirin use and is discussed below under “multipronged cancer interception” [24].

Further research is still required before routinely recommending NSAIDs as cancer interception agents in BE. Future work should focus on detecting differences in the effects of aspirin vs. non-aspirin NSAIDs, the dose needed for cancer-preventive effects, and their point-of-action in gastroesophageal reflux disease-BE-EAC pathogenesis.

### 2.3. Metabolic Pathways

Obesity has notable effects on the insulin signaling, adipokines, and inflammatory pathways which promotes cell proliferation [42]. Multiple studies have shown that obesity and associated metabolic syndromes are significant risk factors for EAC [43,44,45]. An analysis of assorted cancer types revealed that the strongest association between obesity and cancer was for EAC with a relative risk of 4.8 (3.0–7.7) when compared with a normal BMI range of 18.5 to 24.9 [45]. Evidently, BMI alone cannot accurately predict an increased risk of BE and visceral adiposity seems to be a more appropriate marker of risk [43,44,46,47,48]. An important mediator of the association of obesity and EAC is likely the worsening of reflux. The impact of an obesity-induced pro-oncogenic environment, as seen in other cancers, on EAC risk needs further research.

Metformin has been shown to have beneficial effects on insulin resistance and cancer interception [49]. Chak et al. performed a randomized double-blind phase two trial in which 74 participants with a BE of at least 2 cm in size received either metformin (gradually increasing to 2000 mg per day) or a placebo for 12 weeks [50]. There was no difference in the primary endpoint of the percent change in median tissue level of phosphorylated S6 kinase one concentration (pS6K1, mediator of activity of insulin and insulin-like growth factor axis) at 12 weeks between the two groups (1.4% with metformin vs. 14.7% with placebo, one-sided *p*-value of 0.8), even though metformin had a biological effect with reduced levels of serum levels of insulin (median 4.7% with metformin vs. 23.6% increase with placebo). The authors concluded that metformin may not be useful as a direct cancer interception agent to prevent BE progression. However, the results of this “negative” study may be explained by the differential effects of metformin when stratified by BMI. Another trial that evaluated the chemopreventive effect of metformin in stage one lung cancer patients showed the beneficial effect of metformin only in patients with a BMI > 25 (i.e., overweight patients and above), mediated by the down-regulation of multiple checkpoint inhibitor genes, with a detrimental effect in patients with a BMI < 25 [51]. Chak et al.’s study population was stratified by BMI (≥30 vs. <30), placing overweight patients and obese patients in two comparative groups. This could have obscured any potential beneficial effect of metformin. Future trials evaluating metabolic modulators for BE cancer interception should consider adopting a lower cut-off value for BMI to permit appropriate comparisons.

Adipose tissue is now regarded as one of the largest endocrine organs of the body [52]. It secretes metabolically active molecules known as adipokines, which are, in general, pro-inflammatory and promote insulin resistance [53]. Leptin, the classic adipokine, has serum levels that are proportionate to body fat mass [54] and has been demonstrated to be an independent risk factor for the development of BE [55,56,57,58] as well as for progression to EAC [59]. The pro-oncogenic effects of leptin include the modulation of cellular radiosensitivity [60], angiogenesis and lymphangiogenesis [61], and chemoresistance [62]. Studies on OE33 EAC cell lines have shown that leptin positively influences cellular proliferation as well as the malignant potential of these cells (increased migration and invasion) [63,64,65]. Leptin-mediated signaling begins by the binding of leptin to the leptin receptor (ObR), phosphorylation of Janus kinase 2 (JAK2), and then subsequent cascade signaling involving the activation of STAT3, mitogen-activated protein kinases (MAPK), and PI3-kinases/Akt [64] (Figure 2).

Adiponectin, despite being an adipokine, is fundamentally anti-inflammatory, insulin-sensitizing and its secretion from adipocytes varies inversely with body mass, essentially exerting effects opposite to leptin [66,67,68]. It inhibits the leptin-induced cellular proliferation, migration, and invasion of OE33 EAC cell lines in a concentration-dependent manner [63,69]. The proposed signaling pathway is through adiponectin binding to its receptor (AdipoR1) and upregulating the activity and expression of protein tyrosine phosphatase 1B (PTP1B), which is a physiological inhibitor of leptin signaling [63] (Figure 2). Yet, study results for the association of plasma adiponectin with the risk of BE are in disagreement: some studies reported an inverse association [70,71,72], another study detected a statistically significant inverse association only in males but not in females [73], other studies could not find any significant association [55,56], and yet another study found higher adiponectin levels to increase rather than decrease the risk of BE [74]. These discrepancies between in vitro and human studies are possibly due to the different forms (high- vs. low-molecular weight forms of adiponectin; full length vs. globular adiponectin) [75,76,77] and should be taken into consideration when designing studies.

A variety of agents possess the ability to increase serum adiponectin levels: PPAR-γ agonists [68], PPAR-α agonists [78], renin-angiotensin system inhibitors [78], calcium channel modulators [78], and beta-receptor agonists [78], as well as the phytochemical catechin [79]. Osmotin, a plant protein, has been discovered to possess structural and physiological similarities to adiponectin [80]. Data are unfortunately limited and their efficacy in the context of esophageal diseases is underexplored.

Atorvastatin reduced leptin-induced Akt signaling in OE33 cells, presenting a viable option for leptin modulation and cancer interception [81] (Figure 2). The preclinical development of adiponectin-analogs [82,83] and leptin-receptor antagonists [84] is underway; however, more research is required before they can be put to clinical use.

### 2.4. Statins

Statins are another promising class of agents that can have profound metabolic effects and might promote cancer prevention (Figure 1 and Figure 2). Statins inhibit 3-hydroxy-3-methylglutaryl coenzyme A (HMG-CoA) reductase and can affect multiple processes related to cancer progression including proliferation, apoptosis, angiogenesis, and immune modulation [85]. Epidemiologic support for the use of statins in patients with BE comes from a meta-analysis of 9285 cases of esophageal cancer (combined the histologic subtypes of squamous and adenocarcinoma) among nearly 1 million participants and showed a 28.0% risk reduction [86]. More importantly, in a subset analysis of five studies that focused on patients with BE, the risk reduction was 41.0% (adjusted OR, 0.59, 95% CI, 0.45–0.78) [86]. Another study aimed to demonstrate the effect of statins on cell growth and metastatic potential via the downregulation of ICAM (a marker of metastasis) on the cell surface and reduced activation of its mediator, NF-kappa B. Simvastatin treatment of an EAC cell line Flo-1 led to a dose-dependent increase in apoptosis, a decrease in proliferation, decreased total cellular ICAM-1 expression and suppressed NF-kappa B activation. However, atorvastatin showed only mild effects and pravastatin virtually none [87]. Another study that evaluated the in vitro effects of three different statins on two other EAC cell lines, OE33 and BIC-1, showed that all statins had similar effects on reducing cell viability and inhibiting proliferation [88]. It remains to be seen whether the cancer interception effects depend on the specific statin or are a class effect, an important point to carefully consider before planning prospective clinical trials.

### 2.5. Repurposing Cancer Treatment Drugs

One of the new paradigms has been to apply known chemotherapeutic agents (in lower, more tolerable doses) in the field of cancer interception, particularly in high-risk patients. For instance, erlotinib, a tyrosine kinase inhibitor, was tested at half of the usual dose used for cancer chemotherapy in patients with familial adenomatous polyposis but without active cancer. Erlotinib reduced the risk of gastrointestinal polyps by approximately 77% [89]. The expectation is that using lower doses would make the drug more tolerable and shift the risk–benefit profile favorably in patients at high risk for cancer. This approach has not been formally tested for cancer interception in BE, but there is support from in vitro models. An important pre-invasive pro-oncogenic pathway in BE patients with high-grade dysplasia involves p27 phosphorylation by src kinase, which would prevent p27 from exercising its regulatory function [90,91,92]. Using in vitro models of immortalized BE cell lines established from patient biopsies, the investigators validated the above hypothesis using a small molecule inhibitor of SRC, dasatinib, to restore p27’s regulatory function, thus inhibiting cell proliferation, promoting cell-cycle arrest, and activating apoptosis [93]. These data suggest that repurposing cancer treatment drugs such as dasatinib may be an option for patients with BE and high-grade dysplasia (Figure 2).

An improved understanding of molecular processes that are active in the pre-cancer stage could lead to the translation of drugs that typically belong in the realm of cancer treatment to that of cancer interception. A plausible example is the VEGFR2 antagonist, ramucirumab, which found success in the REGARD trial in prolonging the survival of patients with advanced gastric or gastro-esophageal junctional adenocarcinoma [94]. VEGF signaling seems to play a role in BE-EAC pathogenesis [11]. Though premature, these data beget the question of whether ramucirumab could be used in cancer interception in BE (Figure 2). Molecular phenotyping of EAC cell lines by Fitzgerald and colleagues has provided molecularly targeted candidates that could be tested further for cancer prevention in high-risk BE [95]. Much work remains to be carried out in this exciting field where, conceptually, specific chemotherapeutic drugs could be another option in lieu of endoscopic mucosal resection and ablation of high-risk BE.

### 2.6. Multipronged Chemoprevention

Most invasive cancers are formed when an estimated one to five driver genes mutate [96,97]. Multiple pro-oncogenic pathways are simultaneously active to result in the progression of BE to cancer. It then follows that multipronged chemoprevention, to target multiple pathways simultaneously, could be more effective than single-agent prevention.

In the meta-analysis referred to above, a subgroup analysis showed that the combination of statins with NSAIDs/aspirin reduced the risk of EAC by 72.0% (adjusted OR: 0.28, 95% CI: 0.14–0.56), which was higher than that seen with individual medications [82]. Cost-effective modeling analysis suggests that the combination of statins with aspirin had better incremental cost-effectiveness ratios (ICER) of USD 96,000/QALY (quality-adjusted life years), with a willingness-to-pay threshold of USD 100,000/QALY, in high-risk patients when annual progression rates were 0.5%/year [98].

As discussed above, the nature of acid- and bile-induced esophageal injury may be inflammatory rather than chemical. Based on these data, one of the largest trials of cancer interception in BE, the AspECT trial, was conducted. Using a two-by-two factorial design, the investigators randomized nearly 2500 patients with biopsy-proven BE to low- (20 mg QD of esomeprazole) or high-dose PPI (40 mg BID of esomeprazole), with or without aspirin (300 mg/day) [24]. The primary composite endpoint included overall mortality or incident cancer or incident high-grade dysplasia, and 313 such events occurred after a mean follow-up of ~9 years. The patients who received high-dose PPI fared better than those who received low-dose PPI with a time ratio of 1.27 (95% CI: 1.01–1.58, *p* = 0.039). In practical terms, patients in the high-dose arm had a delayed onset of the composite endpoint by approximately two years. Although there was no effect of aspirin in the primary analysis, there seemed to be a benefit when NSAID use was censored. If cancer-related outcomes were scrutinized, there seems to be no effect of either high-dose PPI or aspirin. Overall outcomes were the best in patients who took high-dose PPI and aspirin, but there was no clear measurable benefit of the combination on cancer-related outcomes in patients with BE. To put the results in perspective, this trial assumed an equal risk of progression for all patients with BE. As the progression risk can be variable, there is potentially a subgroup of patients that is at higher risk for progression and should be the focus of future cancer interception trials with the combination of PPI and aspirin.

### 2.7. Miscellaneous

Table 1 lists various medications that have been tried for chemoprevention in BE. A front-runner has not emerged. A newly recognized mediator of injury in BE could be HIF 2 alpha [14]. HIF alpha inhibitors are now available and, although this may be some time in the future, these drugs could have a role in BE chemoprevention [15] (Figure 1). 

### 2.8. Post-Ablation Recurrences of Barrett’s Esophagus

Approaches to reduce post-ablation BE will be important to the future of cancer interception in BE as endoscopic ablative therapy is widely practiced. The goal of endoscopic therapy is to not only achieve the complete eradication of dysplasia but also the complete eradication of intestinal metaplasia to reduce the risk of metachronous lesions. For BE with irregular or nodular esophageal mucosa, the resection of the area with either endoscopic mucosal resection or endoscopic submucosal dissection is performed to remove the lesion and exclude the invasive cancer that might be an indication for esophagectomy. Endoscopic ablative therapy is recommended for flat BE in most patients that have a history of dysplastic BE or high-risk features for the development of dysplasia with a goal of achieving the complete eradication of intestinal metaplasia. Radiofrequency ablation is the most commonly utilized modality due to its widespread availability and based on its demonstrable efficacy and safety [99,100,101,102].

A major concern with radiofrequency ablation, however, is the significant risk of recurrence of BE, even after the complete eradication of intestinal metaplasia. Risk factors include increasing age, BE segment length [103], and pre-treatment dysplasia [104]. Estimates of incidence rates have been variable, ranging anywhere from 8.6% to 10.8% per patient-year [103,105,106,107]. Although the reasons for the recurrence of BE remain unclear, pathways such as Hedgehog, originally functioning to regulate progenitor/stem cells and initiate BE via transcommitment [108], are assumed to remain active and cause recurrent BE. Acid and bile reflux have been linked to this cellular reprogramming [109], thus PPIs are commonly prescribed for preventing recurrent BE. By creating an acid-suppressed environment with high-dose PPIs, healing following endoscopic eradication occurs by the regeneration of the native squamous epithelium (rather than columnar BE epithelium) from progenitor/stem cells. Komanduri et al. [110] corroborated this theory of a structured reflux control protocol after ablation, lowering the recurrence of IM significantly when compared to historical controls.

While PPIs are promising, targeting oncogenic signaling pathways offers an alternative mode of action for preventing recurrent BE. Itraconazole, an anti-fungal drug, has been shown to inhibit Hh signaling, among other pathways [111]. Several clinical trials have successfully showcased the antitumor capabilities of this drug, including in patients of basal cell carcinoma [112], prostate cancer [113], and non-small-cell lung cancer [114]. Given the critical role of the Hedgehog pathway in BE and its recurrence, itraconazole could theoretically be repurposed for cancer interception in this context (Figure 2). Indeed, multiple in vitro and in vivo studies have yielded positive results. In one study [115] using a surgical rat reflux model, the incidence of BE and progression to EAC was compared between the itraconazole-treated group and the control group. They found that, while the proportion of animals developing BE remained high and almost equal between the two arms, fewer rats progressed to EAC when administered itraconazole. The absence of an effect on BE incidence could be due to itraconazole being administered six months after the surgery after BE was likely established. Chen et al. [116] found itraconazole, through AMPK activation, could inhibit cancer cell growth in both established (Eca-109 and TE-1 cell lines) and primary human esophageal cancer cells. Another study [117] revealed a direct correlation between the decreased growth of OE-33-derived flank xenografts in mice and the attenuated pathway activity of Hedgehog, AKT, and VEGFR2. The same group conducted a window-of-opportunity early phase I study to test the viability of neoadjuvant itraconazole in patients with resectable esophageal carcinoma. A biopsy of the tumors showed a decrease in HER2-AKT signaling, attributable to itraconazole. In the face of such compelling evidence, our group is involved in an ongoing proof-of-principle study [118] to generate pilot data to design trials to test the hypothesis that itraconazole can reduce the risk of BE recurrence after endoscopic therapy in patients with high-risk BE. 

Subsquamous intestinal metaplasia (SSIM), essentially “buried” islands of metaplastic tissue, could underlie Barrett’s recurrence post-ablation [119,120,121]. Epithelial-mesenchymal plasticity in response to acid and bile induced injury has been purported to be the culprit [11,122]. This is still a young and under-characterized topic but SSIMs may act as a future target for cancer interception trials.

**Table 1 ijms-24-11318-t001:** Summary of clinical trials of cancer interception in Barrett’s esophagus ^1^.

Publication	Type of Study	Study Population	Chemopreventive Agent(s)	Comparison Groups	Endpoints/ Outcomes	Main Results
Abrams et al. (2021) [123]	Randomized, double-blind, placebo-controlled trial	20 BE patients on continuous PPI therapy	Gastrin/CCK2R antagonist	Netazepide (25 mg/day) vs. placebo	Ki67 density (Ki67 + cells/mm^2^) as a measure of cellular proliferation Gastrin and CgA levels Gene expression analyses	- No difference in change in cellular proliferation when comparing two arms - Netazepide treatment resulted in increased expression of gastric phenotype genes and cancer-associated markers, with decrease in expression of intestinal markers
Valenzano et al. (2020) [124]	Phase I pilot study	10 BE patients	Zinc gluconate	Zinc gluconate (52.8 mg/day) vs. placebo	Zinc-induced transcriptional changes in RNA isolated from Barrett’s biopsy	- Increased anti-inflammatory pattern of gene expression - Decreased expression of mediators of epithelial mesenchymal transition - Upregulation of tumor suppressor genes
Jankowski et al. (2018) [24]	Phase III, randomized prospective 2 × 2 factorial trial	2557 BE patients	PPI, aspirin	High dose esomeprazole (80 mg/day) vs. Low dose esomeprazole (20 mg/day); Aspirin (300/325 mg/day) vs. No aspirin	Composite endpoint of time to all-cause mortality, EA, or HGD (whichever occurs first)	- High dose PPI significantly more effective than low dose PPI (TR 1.27, 95% CI 1.01–1.58, *p* = 0.038) - Largest effect with high dose PPI + aspirin vs. Low dose PPI + no aspirin (TR 1.59, 95% CI 1.00–2.29, *p* = 0.053)
Cummings et al. (2017) [125]	Multicenter, nonrandomized, interventional pilot study	18 BE patients	Vitamin D3	BE cells pre- and post-treatment (Vit D3 50,000 IU/week) vs. normal esophageal cells	Global gene expression, histology, 15-PGDH IHC	- No significant differences
Banerjee et al. (2016) [126]	Open label, single-arm interventional trial	29 BE patients	UDCA	BE cells pre- and post-treatment (UDCA 13–15 mg/kg/day)	Oxidative DNA damage (8-hydroxydeoxyguanosine levels) Cell proliferation (Ki67 expression) Apoptosis (cleaved caspase 3)	- No significant changes in markers of oxidative DNA damage, cell proliferation, and apoptosis
Bratlie et al. (2016) [127]	Prospective, double-blind, triple-arm, randomized trial	30 BE patients	ACE inhibitor, AT1R antagonist	High dose PPI (Esomeprazole 40 mg/day) vs. HD PPI + ACEI (Enalapril 5 mg/day) vs. HD PPI + AT1R blocker (Candesartan 8 mg/day)	Expression of proteins known to be associated with inflammation, proliferation, and cancer development (p53, caspase 3, iNOS)	- Increased expression of iNOS in AT1R blocker group (*p* = 0.033) - Increased p53 expression (*p* = 0.05), decreased AMACR (*p* = 0.017) and caspase 3 (*p* = 0.025) in ACEI group. - Decreased expression of inflammation related factors NF-kappa B and NLRP3 (*p* = 0.043 and *p* = 0.05)
Chak et al. (2015) [50]	Randomized, double-blind, placebo-controlled trial	74 BE patients	Metformin	Metformin (increasing to 2000 mg/day) vs. placebo	Change in mean pS6K1 level Cellular proliferation (Ki67 assay) Apoptosis (levels of caspase 3)	- No significant differences between the arms
Joe et al. (2015) [128]	Phase 1b Randomized, double-blinded, placebo-controlled dose escalation study	44 BE patients	Polyphenon E	Poly E (200 mg BID/400 mg BID/600 mg BID) vs. Placebo	Esophageal tissue levels of catechins Endoscopic measurement of BE	- Catechin EGCG accumulation in esophageal mucosa - No reduction in length of BE - No significant changes in mucosal protein expression
Peng et al. (2014) [129]		21 BE patients	UDCA	Perfusion of BE cells with DCA and UDCA vs. 24-h pretreatment of BE cells with UDCA followed by perfusion with DCA vs. Pretreatment with oral UDCA (10 mg/kg) followed by perfusion with DCA	Molecular analysis of BE cells	- Both oral UDCA and pretreatment with UDCA increased antioxidant protein expression (GPX1, catalase) thus preventing DNA damage and NK-kappa B activation in Barrett’s cells which have been exposed to toxic bile acids.
Falk et al. (2012) [130]	Phase 2, multicenter, randomized, double-blind, placebo-controlled trial	122 BE patients	Aspirin	Aspirin placebo + PPI vs. Lower dose aspirin + PPI vs. Higher dose aspirin + PPI	Absolute change in mean tissue PGE2 concentration	- Statistically significant difference in post intervention PGE2 values for both lower dose and higher dose aspirin (signed rank test *p* = 0.0004) - Higher dose aspirin + PPI resulted in significant decrease in PGE2 values, as compared to aspirin placebo + PPI (*p* = 0.02)
Rawat et al. (2012) [131]	In vitro and in vivo pilot study	36 BE patients	Curcumin	Curcumin tablet (500 mg/day) Vs. No medication	IL-8 and I-κB gene expression	- No significant difference between both groups
de Bortoli et al. (2011) [132]	Single-center, open-label, randomized, parallel group design trial	77 BE patients	PPI	Esomeprazole (80 mg/day) vs. Pantoprazole (80 mg/day)	Cell proliferation (Ki67 expression) COX-2 expression Apoptosis (TUNEL detection)	- Esomeprazole group demonstrated statistically significant decrease in Ki67 and COX-2 expression and an increase in apoptotic cell death (compared to baseline values) - No significant difference in pantoprazole group
Babar et al. (2010) [133]	Pilot, translational, proof-of-concept trial	25 BE patients on continuous PPI therapy	Vitamin C	BE cells pre- and post-treatment (Redoxon 1000 mg/day)	NF-kappa B activation Cytokine profile (VEGF, IL8, IL1α, IL1β)	- Oral supplementation with Vitamin C resulted in a 10% or greater down-regulation of NF-kappa B in 32% of patients. - Mean levels of cytokines were lower in patients who showed ≥10% down-regulation in NF-kappa B than those who did not.

^1^ This table is not intended to be an exhaustive list of all clinical trials in this field. Trials are listed according to the year in descending order. Legend: CCK2R, Cholecystokinin 2 receptor; PPI, Proton pump inhibitor; EA, Esophageal adenocarcinoma; HGD, High grade dysplasia; TR, Time ratio; 15-PGDH, 15-hydroxyprostaglandin dehydrogenase; IHC, Immunohistochemistry; iNOS, inducible nitric oxide synthase; AMACR, Alpha methylacyl CoA racemase; ACEI, Angiotensin-converting enzyme inhibitor; NLPR3, Nod-like receptor protein 3; pS6K1, Phosphorylated ribosomal protein S6 kinase-1; Poly E, Polyphenon E; EGCG, Epigallocatechin gallate; PGE2, Prostaglandin E2; VEGF, Vascular endothelial growth factor; IL8, Interleukin-8; IL1α, Interleukin-1 alpha; IL1β, Interleukin-1 beta.

## Figures and Tables

**Figure 1 ijms-24-11318-f001:**
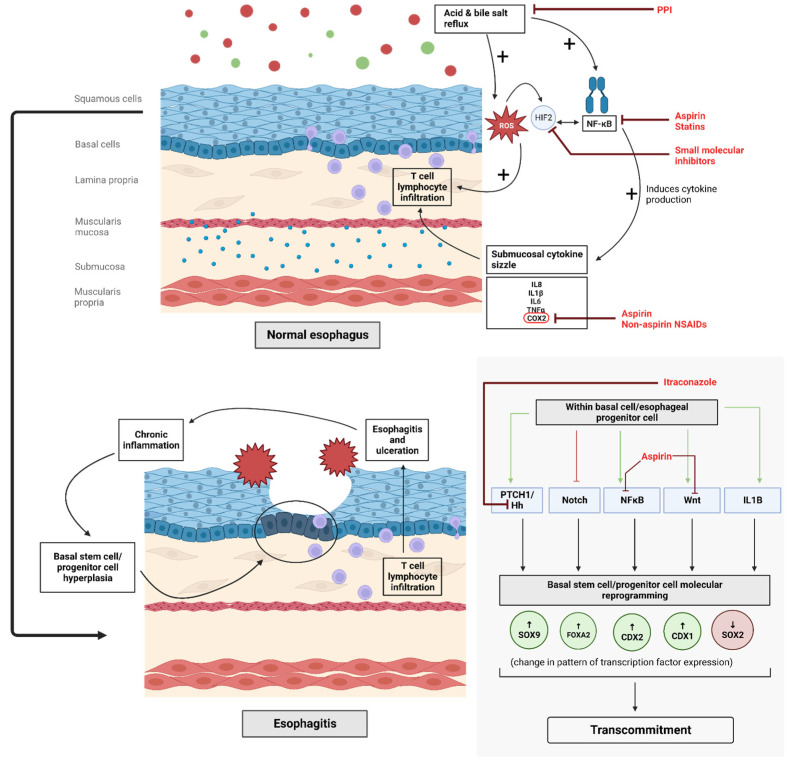
Proposed molecular pathways involved in Barrett’s esophagus initiation. Specific drugs (in red) with their postulated targets are highlighted. PPI—proton pump inhibitor; HIF2—Hypoxia inducible factor 2; IL1β—Interleukin 1 Beta; Hh—Hedgehog; PTCH1—Patched 1; Wnt—Wingless/Integrated; SOX9—SRY-box transcription factor 9; FOXA2—Forkhead Box A2; CDX1—Caudal Type Homeobox 1; CDX2—Caudal Type Homeobox 2; SOX2—Sex-determining region Y-box 2.

**Figure 2 ijms-24-11318-f002:**
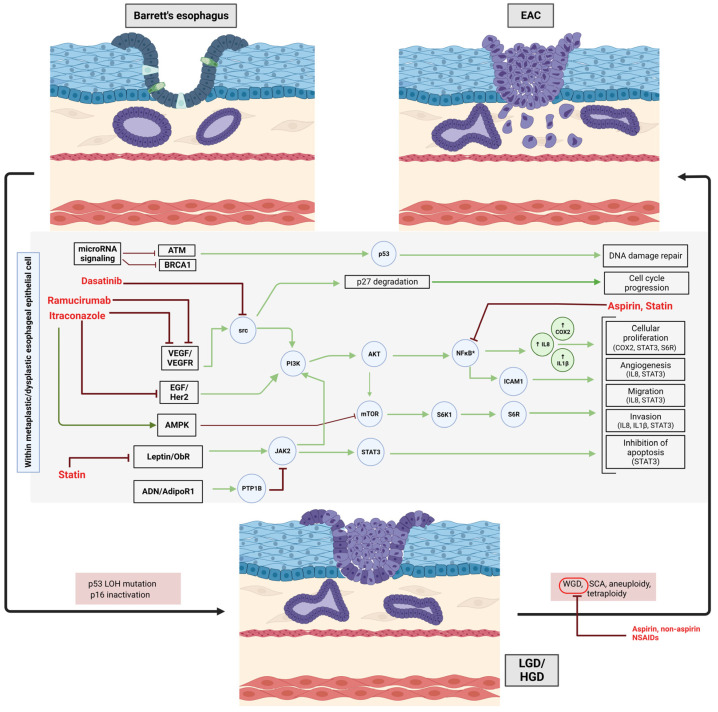
Proposed molecular pathways involved in progression of Barrett’s esophagus to dysplasia to esophageal adenocarcinoma. Specific drugs (in red) with their postulated targets are highlighted. Activation of NFkB mirrored by concomitant increase in COX2, IL8, and IL1β along the spectrum of BE, dysplasia, and EAC. LGD—Low Grade Dysplasia; HGD—High Grade Dysplasia; EAC—Esophageal Adenocarcinoma; ATM—Ataxia telangiectasia mutated; BRCA1—Breast Cancer gene 1; VEGF—Vascular Endothelial Growth Factor; VEGFR—Vascular Endothelial Growth Factor Receptor; EGF—Epidermal Growth Factor; src, Src family protein-tyrosine kinase; PI3K—Phosphoinositide 3-kinase; AKT—Protein Kinase B; NF-κB—Nuclear Factor kappa-light-chain-enhancer of activated B cells; COX-2—Cyclooxygenase-2; IL8—Interleukin-8; IL1B—Interleukin 1 Beta; Her2—Human epidermal growth factor receptor 2; AMPK—Adenosine Monophosphate activated protein kinase; mTOR—Mammalian target of rapamycin; S6K1—Ribosomal Protein S6 Kinase 1; S6R—Ribosomal Protein S6; ObR—Leptin Receptor; Jak2—Janus Kinase 2; STAT3—Signal Transducer and Activator of Transcription 3; AND—Adiponectin; AdipoR1—Adiponectin Receptor 1; PTP1B—Protein Tyrosine Phosphatase 1B; WGD—Whole Genome Doubling; SCA—Somatic Chromosomal Alterations.

## Data Availability

Data sharing not applicable.

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
