# Peer review of "Translating Molecular Biology Discoveries to Develop Targeted Cancer Interception in Barrett’s Esophagus"

_ijms, 2023, doi:10.3390/ijms241411318_

Round 1
Reviewer 1 Report
Translating molecular biology discoveries to develop targeted cancer interception in Barrett’s esophagus is a review paper aimed to the mechanisms of action of various agents, including proton pump inhibitors, non-steroidal anti-inflammatory drugs, metformin, and statins, and explain their roles in cancer interception.
The paper is of good quality and well performed.
It examines the panorama present in the literature on this topic in an extremely exhaustive way.
In my opinion it could be accepted for publication.
Congratulations to the Authors
No issues detected
Author Response
We thank the reviewer for their kind words and feedback to improve the quality of our manuscript. Their comments were helpful and constructive.
Sincerely,
Ajay Bansal, MD
Reviewer 2 Report
The manuscript by Samaddar and coworkers is well written. The table and the figures are accurately designed. The content is interesting and the manuscript can be suitable for publication after minor revision:
Published data from dietary compounds such as curcumin and polyphenon E in terms of Barrett’s esophagus treatment or prevention might be provided or at least cited and discussed in comparison with the compounds mentioned in this manuscript.
Line 268: Please replace ´´Table´´ by ´´Table 1´´. The same in the header of ´´Table 1´´.
Figure 2: There are different forms of ´´src´´ and ´´src kinase´´ in this figure. Please explain the difference between both or make conform. Please clarify it in the caption, too.
There is no Discussion section. Please check again if this complies with the journal guidelines.
References need to be adjusted to the journal style.
Reviewer 3 Report
Interesting work on an extremely complex issue. Controversies are numerous despite many studies and research on the issue. Some comments that can be considered questions for the authors.
Line 25
It is not obesity that is the etiological mechanism, but GERD. Obesity is just one of the factors that can favor or induce reflux
Line 41
PPIs could not play a role in cancer interception in BE or if there is a role, it is only a marginal one. About 30-40% of reflux pathology is non-acidic and it is precisely in these cases that most adenocarcinomas appear.
Acid-suppressive drug therapy only addresses symptoms and cellular changes in the esophagus caused by acid. Acid suppression does not have any positive effect on the LES, the defective function of which is the cause of reflux. In addition, PPIs do not reduce reflux, having no effect on physiological anti-reflux mechanisms (e.g. LES), PPIs only change the quality of reflux. Failure of PPIs to control symptoms is inevitable in those patients whose LES damage is destined to progress to a high degree of severity during their lifetime.
Line 45
Exposure to gastric acid juice induced increase in permeability of the squamous epithelium allowing the entry of a variety of molecules which are contained in gastric juice. Acid-reducing therapy therefore converts strong acid reflux to weak acid reflux exposure of the thoracic esophageal epithelium. This is clear. By consequence, when acid suppression maintains an intragastric pH > 4 for >12 h/day, the likelihood of healing erosive esophagitis and preventing recurrence of erosive esophagitis are high. But alkalinization of gastric juice probably has an indirect effect in increasing cancer risk.
There is now a strong theoretical reason for the epidemiologically demonstrated association between the use of acid-suppressive drugs and esophageal adenocarcinoma. The action of PPIs and other alkalinizing agents is to increase the pool of patients with Barrett esophagus. Why? The cellular change that is most affected by alkalinization of gastric juice is the development of intestinal metaplasia (i.e., Barrett esophagus) in nonintestinalized cardiac epithelium in the esophagus that has resulted from GERD. An unknown but very commonly present molecule in gastric juice must interact with the squamous epithelial cells to produce the genetic switch that causes columnar metaplasia. The carcinogens responsible for reflux-induced carcinogenesis are suspected of being derived from bile acids that reach the stomach via duodenogastric reflux. There is a strong association between adenocarcinoma of the esophagus and the presence of duodenal contents in the esophagus. The metabolism of bile acids in the stomach is highly pH-dependent. In normal patients, the gastric acidity (pH 1–3) causes the bile acids to precipitate and become inactive. In the pH range of 3–5, which is usual in most patients who are on acid-suppressive drug treatment, bile acid metabolism results in the production of soluble, un-ionized bile acid derivatives. These have been shown experimentally to enter esophageal epithelial cells and result in the activation of genetic pathways that are associated with increased cellular proliferation. Alkalinization of the refluxate has a negative effect on the cardiac mucosa, causing it to undergo intestinal metaplasia at an increasing distal point in the esophagus. This, by bringing the target cell ever closer to the effective carcinogen dose in the distal esophagus, promotes cancer. The increased permeability of the squamous epithelium when exposed to gastric juice must therefore be of sufficient severity to allow the critical unknown large molecule in gastric juice to penetrate the epithelium to a depth where it can interact with proliferative cells in the prickle cell or basal layer
Line 117
The link between obesity and esophageal adenocarcinoma is "mediated" by reflux, by the increase and intensification of GERD symptoms. All adenocarcinomas of the esophagus are reflux-induced cancers occurring in the surface epithelium. There is no other known mechanism for adenocarcinoma in the esophagus except rare tumors arising in esophageal glands.
Line 273
At present, there is controversy about whether these patients should be treated with a limited esophagectomy or some other form of ablation (endoscopic mucosal resection, radiofrequency ablation, or photodynamic therapy), or "wait and see" concept (frequent endoscopy and multiple biopsies) . Attention, we are talking here about the imminence of invasive cancer but this is not known with any degree of certainty.
Nonsurgical ablative methods are appropriate only in patients who cannot tolerate surgery; the lack of a specimen for examination precludes establishing whether there was invasion associated with the high-grade dysplasia. The incidence of any invasive carcinoma transforms a patient who could have been treated effectively with a limited esophagectomy or endoscopic mucosal resection to a candidate for more radical surgery with a higher morbidity
